

# Genetic diversity and population structure of *Euscaphis japonica*, a monotypic species

Wei-Hong Sun[1,2,3], De-Qiang Chen[1,2,3], Rebeca Carballar-Lejarazu[4], Yi Yang[1,2,3], Shuang Xiang[1,2,3], Meng-Yuan Qiu[1,2,3] and Shuang-Quan Zou[1,2,3]

[1] College of Forestry, Fujian Agriculture and Forestry University, Fuzhou, China
[2] Fujian Colleges and Universities Engineering Research Institute of Conservation and Utilization of Natural Bioresources, College of Forestry, Fujian Agriculture and Forestry University, Fuzhou, China
[3] Key Laboratory of National Forestry and Grassland Administration for Orchid Conservation and Utilization at College of Landscape Architecture, College of Landscape Architecture, Fujian Agriculture and Forestry University, Fuzhou, China
[4] Department of Microbiology and Molecular Genetics, University of California, Irvine, CA, USA

Corresponding author
Shuang-Quan Zou,
zou@fafu.edu.com

## ABSTRACT

**Background**. Understanding plant genetic diversity is important for effective conservation and utilization of genetic resources. *Euscaphis japonica* (Thunb.) Dippel, is a monotypic species with high phenotypic diversity, narrow distribution, and small population size. In this study, we estimated the genetic diversity and population structure of *E. japonica* using nine natural populations and inter-simple sequence repeat (ISSR) markers. Our results could provide a theoretical reference for future conservation and utilization of *E. japonica*.

**Results**. We obtained a total of 122 DNA bands, of which 121 (99.18%) were polymorphic. The average number of effective alleles ($Ne = 1.4975$), Nei's gene diversity index ($H = 0.3016$), and Shannon's information index ($I = 0.4630$) revealed that *E. japonica* possessed a high level of genetic diversity. We observed that *E. japonica* consisted of both deciduous and evergreen populations. UPGMA tree showed that the evergreen and deciduous *E. japonica* form a sister group. There is little genetic differentiation among geographic populations based on STRUCTURE analysis. The Dice's similarity coefficient between the deciduous and evergreen populations was low, and the *Fst* value was high, indicating that these two types of groups have high degree of differentiation.

**Conclusion**. Rich genetic diversity has been found in *E. japonica*, deciduous *E. japonica* and evergreen *E. japonica* populations, and genetic variation mainly exists within the population. The low-frequency gene exchange between deciduous and evergreen populations may be the result of the differentiation of deciduous and evergreen populations. We suggest that in-situ protection, seed collection, and vegetative propagation could be the methods for maintenance and conservation of *E. japonica* populations.

## INTRODUCTION

The genetic diversity of woody species is of great significance to their survival and persistence (*Soejima, Maki & Ueda, 2002*). Woody species with high genetic diversity could hold a greater adaptive capacity and are able to adapt to survive in changing environments and under poor conditions (*Gaafar, AI-Qurainy & Khan, 2014*; *Eriksson, Namkoong & Roberds, 1995*; *Hedrick, 2004*). The fecundity and gene dispersal of woody species have been observed to shape their genetic diversity patterns (*Mitchell-Olds, Willis & Goldstein, 2007*). Species with weak regenerating abilities have lower genetic variation and less adaptive flexibility (*Wang et al., 2011*). Habitat fragmentation has an important impact on the demographic and genetic aspects of plant populations. Habitat fragmentation aggravates genetic erosion, and eventually leads to decreased individual fitness, thereby inhibiting population persistence (*White et al., 2020*; *Yang et al., 2016*). Besides, the loss of allele richness caused by decreased heterozygosity may reduce the opportunities for future adaptation of the population (*White et al., 2020*). Geographic range is also a contributing factor to a species' level of genetic diversity, as widely distributed species usually exhibit a higher level of diversity, while endemic woody species often show lower genetic diversity (*Wei et al., 2012*; *Allly, EIKassaby & Ritand, 2000*; *Luan, Chiang & vs, 2006*). Therefore, having a basic understanding of a species' genetic variation is key to developing effective conservation strategies, especially for endangered species or species with a small population (*Booy et al., 2000*; *Vicente et al., 2011*).

*Euscaphis japonica* (Thunb.) Dippel, belongs to the Staphyleaceae family, is monotypic, and only distributed in southern China, Japan, and Korea (*Li, Cai & Wen, 2008*; *Cheng et al., 2010*). According to Flora of China records, *E. japonica* is a deciduous tree or shrub with odd-pinnate leaves, papery leaflets, sparsely serrulate margins with glandular teeth, and a soft, red, leathery pericarp with irregular ribs (*Li, Cai & Wen, 2008*). Previous studies have found that *E. japonica* exhibits significant phenotypic differences at different altitudes, and it can be classified as deciduous or evergreen based on its phenotypic markers (*Sun et al., 2019*). The deciduous *E. japonica* was characterized by its papery leaflets, serrulate margins, and prominent epidermis ribs, and the evergreen *E. japonica* was characterized by its membranous leaflets with obtuse, serrate margins, and its inconspicuous fruit epidermis rib (*Sun et al., 2019*). However, evaluating genetic variations across different populations using morphological characters is difficult because plant morphology varies under different growing conditions (*Wang et al., 2009*).

Furthermore, *E. japonica* is an excellent ornamental tree species for its butterfly-shaped fruit and red pericarp, and is currently cultivated on a large scale as an ornamental and medicinal tree species in Jiangxi and Fuzhou in China. It is also an important medicinal material in China since ancient times for the treatment of colds and coughs (*Liang et al., 2019*; *Sun et al., 2019*). Due to human interference and unreasonable picking of fruits, leaves, and branches for medicinal materials, some habitats of the natural *E. japonica* population have been largely fragmented. Whereas, there is short of a related research on *E. japonica*. A detailed study of its genetic diversity is therefore necessary to develop a

strategy that can manage and maintain resilient, productive, and sustainable *E. japonica* forests (*Iddrisu & Ritland, 2005*).

DNA-based molecular markers are not affected by environmental or physiological factors, making them suitable for estimating genetic diversity across plant species and populations (*Mei et al., 2017*; *Tanya et al., 2011*; *Noormohammadi et al., 2013*; *Kaya, 2015*). Inter-simple sequence repeat (ISSR) marker is a type of molecular markers based on inter-tandem repeats of short DNA sequences, which can determine intramolecular and intergenomic diversity by simultaneously revealing variations in unique regions of several loci in the genome. Because they provide simple, quick, efficient, and reproducible markers that can detect high levels of polymorphism (*Reddy, Sarla & Siddiq, 2002*; *Rakoczy-Trojanowska & Bolibok, 2004*), ISSR markers have been used to analyze the genetic diversity of germplasm collections and to identify genotypes in studies on wild hawthorn (*Sheng et al., 2017*), *Magnolia wufengensis* (*Chen et al., 2014*), *Bergenia ciliata* (*Tiwari et al., 2015*), and *Sindora glabra* (*Yang et al., 2016*). In this study, we collected samples from 83 *E. japonica* individuals across nine populations. We analyzed the genetic variation and population structure of *E. japonica* populations using ISSR markers. We aimed to determine the genetic variations within *E. japonica* populations and across different populations, reveal *E. japonica*'s genetic structure, and recommend future management and conservation strategies.

## MATERIALS & METHODS

### Plant material

Based on the samples collected from the previous phenotypic diversity of *E. japonica* (*Sun et al., 2019*), we have expanded the scope of field investigation and collected more samples from August 2016 to March 2017. During field observation, we found that *E. japonica* had two major categories, deciduous and evergreen, and we sampled these two types separately. All observed individuals were sampled, and the latitude, longitude, altitude, and specific habitats were recorded. We collected 83 samples across nine populations in Wuyi Mountain (WYC1, WYC2, WYC3, WYC4, and WYL), Taimu Mountain (TML), Daiyun Mountain (DYC and DYL), and Xishui National Nature Reserve (ZYL) (Fig. 1). The Wuyi Mountain range is about 550 kilometers long from the northeast to southwest, so five populations were collected. We collected and dried 10 to 15 fresh leaves from each sample using silica gel in sealed bags (*Chase & Hills, 1991*). The dry leaves were stored at 4 °C in a refrigerator to be used for ISSR analysis. The characteristics of the different population localities and morphologies are shown in Table 1.

### DNA isolation and purification

We extracted genomic DNA from the *E. japonica* leaves using a modified acetyl trimethyl ammonium bromide method (*Uddin et al., 2014*). Fresh leaves (50 mg) without veins were ground into a fine powder using liquid nitrogen and a mortar. We then quickly transferred the powder to 2-mL microcentrifuge tubes with 700 µL of extraction buffer (1.4 M NaCl, 100 mM Tris HCl (pH 8.0), 20 mM EDTA, 1% PVP, and 1% β-mercaptoethanol) that had been preheated at 65 °C. This mixture was incubated at 65 °C for 40 min with gentle
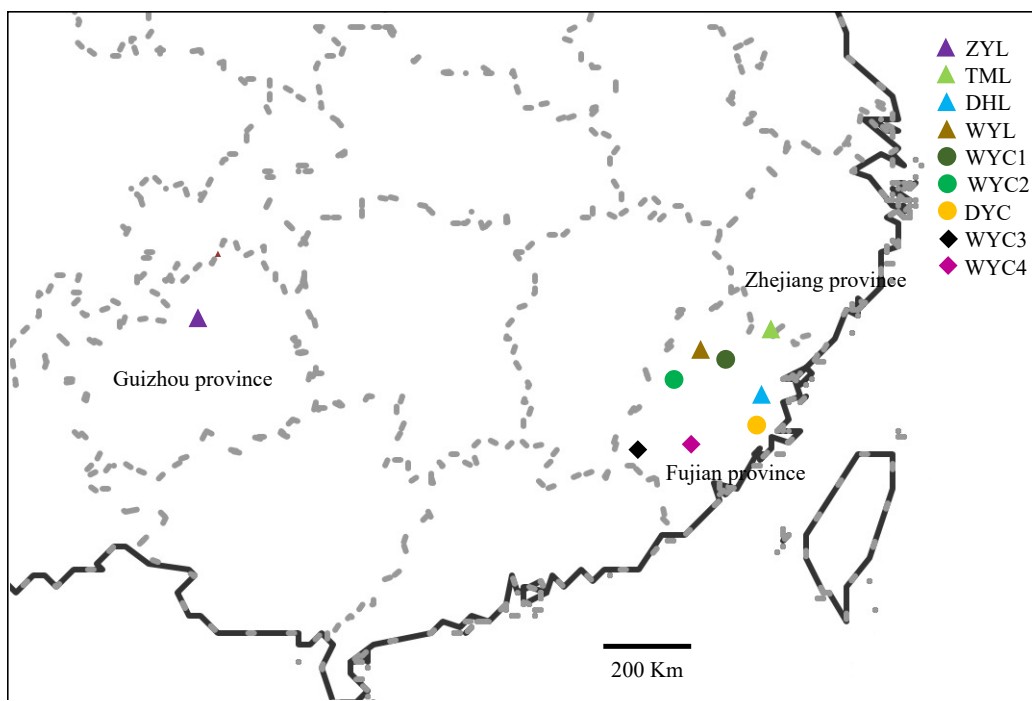

**Figure 1  Map showing the location of the sampled *E. japonica* populations.**

shaking. After cooling for 10 min, we added chloroform: isopentyl alcohol mixture (24:1, equal volume: 700 µL) and centrifuged it at 12,000 rpm for 10 min. The supernatant was transferred into a clean 2-mL microcentrifuge tube, and two volumes of absolute ethanol were added and gently mixed using inversion. The DNA pellet was washed twice with 70% (v/v) ethanol and dissolved in TE buffer (pH 8.0) after air drying. We removed RNA contamination by adding 2 µL of RNase (10 mg/mL) to each tube, incubated the tubes for 30 min at 37 °C, and then added 400 µL of chloroform: isopentyl alcohol (24:1). Samples were centrifuged at 12,000 rpm for 10 min at 4 °C and the aqueous layer was transferred into a new tube. We added two volumes of chilled ethanol and 1% NaAc (3 mol/L, pH 5.2) to precipitate the DNA. The DNA was washed with 70% ethanol and dissolved in 100 µL of TE buffer after air drying. Each sample was diluted to 80 ng/mL using TE buffer and was stored at 4 °C for further ISSR analysis.

We measured the concentration of recovered DNA using NanoDrop™ spectrophotometry (Thermo Fisher Scientific, Waltham, MA, USA) at 260 nm and 280 nm, and verified the purity of the DNA using 0.8% (w/v) agarose gel. SYNGENE (BioRad, Hercules, CA, USA) was used to visualize and photograph the gel under a UV transilluminator.

## ISSR amplification

PCRs contained 1 µL of 80 ng template DNA, 1 µL of 0.5 µmol/L primer, 0.4 µL of 0.2 µmol/L dNTP, 2 µL of 2.5 mmol/L $MgCl_2$, 0.3 µL of 1.5 U Taq enzyme, 2 µL of 10× PCR buffer, and sterile distilled water, for a final volume of 20 µL. ISSR primers were synthesized by Sangon Bioengineering (Shanghai, China) and all reagents were purchased

Sun et al. (2021), *PeerJ*, DOI 10.7717/peerj.12024

**Table 1  The codes, locations, phenotypic character and habitat of populations of *E. japonica*.**

| Location | population (size) | code | E | N | A (m) | Leaf characters | Fruit characters |
|---|---|---|---|---|---|---|---|
| Taimu Mountain, the border of Fujian and Zhejiang Provinces | TML (6) | TML1 ∼ TML6 | 120°08′ | 27°23′ | 540 | Paper, margin sparsely serrulate | pericarp softly leathery, red-brown with irregular ribs |
| Daiyun Mountain, Quanzhou City, Fujian Province, | DYL (7) | DYL1∼ DYL7 | 118°53′ | 26°07′ | 946 | Paper, margin sparsely serrulate | pericarp softly leathery, red-brown with irregular ribs |
| | DYC (3) | DYC1∼ DYC3 | 118°27′ | 25°42′ | 397 | Membrane, margin blunt serrations | Pericarp leathery, without irregular ribs |
| | WYC1 (6) | WYC1-1 ∼ WYC1-6 | 118°18′ | 27°24′ | 181 | Membrane, margin blunt serrations | Pericarp leathery, without irregular ribs |
| | WYC2 (21) | WYC2-1 ∼ WYC2-21 | 117°14′ | 27°03′ | 314 | Membrane, margin blunt serrations | Pericarp leathery, without irregular ribs |
| Wuyi Mountain, the border of Sanming and Nanping Citys, Fujian Province | WYC3 (13) | WYC3-1 ∼ WYC3-13 | 116°48′ | 25°50′ | 340 | Membrane, margin blunt serrations | Pericarp leathery, without irregular ribs |
| | WYC4 (13) | WYC4-1 ∼ WYC4-13 | 114°55′ | 25°23′ | 151 | Membrane, margin blunt serrations | Pericarp leathery, without irregular ribs |
| | WYL (5) | WYL1 ∼ WYL5 | 118°02′ | 27°26′ | 508 | Paper, margin sparsely serrulate | pericarp softly leathery, red-brown with irregular ribs |
| Xishui National Reserve, Zunyi, City, Guizhou Province | ZYL (9) | ZYL1∼ ZYL9 | 106°47′ | 28°49′ | 918 | Paper, margin sparsely serrulate | pericarp softly leathery, red-brown with irregular ribs |

**Notes.**

N, north latitude; E, west longitude; A, altitude (m).

Population name ending in 'L' represents the deciduous population, and population name ending in 'C' represents the evergreen population.

**Table 2  ISSR primer used for ISSR-PCR amplification and their amplification results.**

| Primer | Sequence | %GC | No. of bands | PIC | Nm |
|--------|----------|-----|--------------|-----|-----|
| UBC807 | AGAGAGAGAGAGAG AGT | 47.1 | 12 | 0.57 | 1.4439 |
| UBC808 | AGAGAGAGAGAGAGAGC | 52.9 | 14 | 0.50 | 1.5361 |
| UBC809 | AGAGAGAGAGAGAGAGG | 52.9 | 10 | 0.62 | 1.8155 |
| UBC816 | CACACACACACACACAT | 53 | 10 | 0.61 | 2.3000 |
| UBC818 | CACACACACACACACAG | 52.9 | 9 | 0.55 | 0.8754 |
| UBC825 | ACACACACACACACACT | 47.1 | 9 | 0.58 | 3.3207 |
| UBC826 | ACACACACACACACACC | 52.9 | 9 | 0.66 | 0.5512 |
| UBC827 | ACACACACACACACACG | 52.9 | 10 | 0.52 | 1.5393 |
| UBC856 | ACACACACACACACYA | 47.2 | 10 | 0.48 | 2.2742 |
| UBC861 | ACCACCACCACCACCACC | 66.7 | 12 | 0.50 | 1.6249 |
| UBC862 | AGCAGCAGCAGCAGCAGC | 66.7 | 11 | 0.61 | 1.2078 |
| UBC890 | VHVGTGTGTGTGTGTGT | 51.0 | 6 | 0.51 | 3.6068 |
| Mean | | | 10.17 | 0.56 | 0.7804 |

from Solarbio (Beijing, China). We conducted PCR amplification using a Veriti$^{TM}$ 96-Well Thermal Cycler (Applied Biosystems, Foster City, CA, USA) as follows: initial denaturation at 95 °C for 5 min, 35 cycles at 94 °C for 1 min, at 55 °C for 30 s, and 72 °C for 1 min; final extension at 72 °C for 7 min, and the tubes were subsequently maintained at 4 °C before analysis. All ISSR primers had been initially tested, and 12 primers amplified DNA with polymorphic bands (Table 2). PCR products were analyzed on a 1.2% (w/v) agarose gel in 1 × TBE buffer and were visualized using the Gel Doc system (BioRad).

## Statistical analysis

We scored amplification based on the amplicon bands from the gel photographs. The presence of bands at an amplicon level was scored as 1 and the absence was scored as 0, which we used to calculate the raw data of the 0–1 matrix (*Uddin et al., 2014*). We used POPGENE 1.3.1 to analyze various genetic parameters, such as the number of alleles ($Na$), the number of effective alleles ($Ne$), Nei's genetic diversity ($H$), Shannon's information index ($I$), the percentage of polymorphic loci ($PPB$), the level of gene flow ($Nm$), and the Dice's similarity coefficient (GS) (*Yeh et al., 1997*). We calculated polymorphism information content (PIC) using the following formula: PIC = 2 Pi (1–Pi), where Pi is the frequency of polymorphic band occurrence in different primers (*Khaleghi et al., 2017*). The unweighted pair group method with the arithmetic average (UPGMA) and NTSYS 2.1 were used for dendrogram construction. To verify the UPGMA clustering results, we performed a cophenetic correlation analysis using NTSYS 2.1. Principal component analysis (PCA) was performed to show the *E. japonica* multiple dimensional distributions in a scatter plot (NTSYS 2.1) (*Wang et al., 2009*). We used GenAlex 6.5 to calculate the variance components within and between populations (*Peakall & Smouse, 2012*). The genetic structure of the *E. japonica* population was determined using STRUCTURE 2.3.1 based on Bayesian model-based clustering (*Pritchard, Stephens & Donnelly, 2000*). The procedure was carried out by selecting the correlated allele frequencies among populations and the admixture
model. The presumed populations ($K$) denoted from 1 to 13 and estimated 20 independent runs for each $K$. The operating parameter was a burn-in period of 100,000 and 100,000 Markov Chain Monte Carlo replicates after burn-in. The optimal number of clusters was identified using Structure Harvester (http://taylor0.biology.ucla.edu/struct_harvest/) (*Earl & vonHoldt, 2012*).

## RESULTS

### ISSR polymorphism and genetic diversity

We amplified a total of 122 identifiable bands, 121 of which were polymorphic bands. The polymorphic ratio (*PPB*) was 99.18%. The size of the PCR products ranged from 250 to 2,000 bp. The largest number of bands (14) was obtained using the UBC 808 primer, while only six bands were obtained using the UBC 890 primer. The highest *PIC* value was 0.62 (UBC 809), the lowest *PIC* value was 0.48 (UBC 856), and the mean *PIC* per primer was 0.56 (Table 2). The high *PPB* and *PIC* values suggest that *E. japonica* has abounding genetic information, and that ISSR markers are appropriate to use in the genetic diversity analysis of *E. japonica* populations

Table 3 shows the statistical analysis of the *E. japonica* genetic parameters when using GenAIEx 6.5. At the species level, the average *Ne* was 1.4974, *H* was 0.3016, and *I* was 0.4630. At the population level, the ranges of *H* and *I* across different populations were 0.0994–0.3688 and 0.1594–0.3971, respectively. Five populations had *H* values greater than 0.2: WYC2 (0.2607), WYC3 (0.2512), WYL (0.2360), DYL (0.2301), and WYC4 (0.2141). Four populations had *I* values greater than 0.3: WYC2 (0.3971), WYC3 (0.3751), ZYL (0.3548), and DYL (0.3471). The five populations with the largest *Ne* values were WYC2 (1.4090), WYC3 (1.4276), ZYL (1.4034), DYL (1.3857), and WYC4 (1.3610). The five populations with the largest population polymorphic rates (*PPB*) were WYC (86.07%), WYC4 (72.13%), WYC3 (70.49%), ZYL (68.85%), and DYL (67.21%). Considering each genetic diversity index collectively, including the *H*, *I*, and *Ne* values, the WYC2 population had the highest genetic diversity, followed by WYC3 and WYC4.

### Genetic structure

The UPGMA tree, based on individual samples, consisted of four major clusters. Most ZYL population samples and all WYL samples were grouped into cluster A. The remaining ZYL samples, all TML samples, and DYL samples were grouped into cluster B. Cluster C contained WYC3 and WYC4 samples. Cluster D included WYC1, WYC2, and DYC samples. Clusters A and B consisted of deciduous *E. japonica*, and clusters C and D consisted of evergreen *E. japonica* (Fig. 2A). We based the PCA on genetic similarities to determine the genetic relationships between *E. japonica* populations and to create a two-dimensional display of these relationships. The PCA of the cumulative data grouped the nine populations into clusters I, II, and III, where cluster I included deciduous *E. japonica* and clusters II and III included evergreen *E. japonica*. Cluster II contained WYC1, WYC2, and DYC samples, and cluster III included WYC3 and WYC4 samples (Fig. 2B). The STRUCTURE analysis results indicated that the number of optimal clusters was four (Figs. 2C and 2D) based on the maximum delta $K = 4$. Therefore, we divided the populations into four clusters

**Table 3  Genetic diversity from *E. japonica* populations.**

| Population | *Na* | *Ne* | *I* | *H* | *He* | *uHe* | PPB (%) |
|---|---|---|---|---|---|---|---|
| TML | 1.3197 | 1.1735 | 0.1594 | 0.1046 | 0.1046 | 0.1141 | 31.97 |
| DYL | 1.4344 | 1.3857 | 0.3471 | 0.2301 | 0.2301 | 0.2422 | 67.21 |
| DYC | 1.2130 | 1.1244 | 0.1115 | 0.0738 | 0.0738 | 0.0843 | 21.31 |
| WYC1 | 1.2787 | 1.3483 | 0.2982 | 0.2003 | 0.2003 | 0.2185 | 55.74 |
| WYC2 | 1.7705 | 1.4390 | 0.3971 | 0.2607 | 0.2607 | 0.2670 | 86.07 |
| WYC3 | 1.4918 | 1.4276 | 0.3751 | 0.2512 | 0.2512 | 0.2612 | 70.49 |
| WYC4 | 1.5820 | 1.3610 | 0.3278 | 0.2141 | 0.2141 | 0.2220 | 72.13 |
| WYL | 0.7131 | 1.1448 | 0.1174 | 0.0803 | 0.0803 | 0.0892 | 20.49 |
| ZYL | 1.4426 | 1.4034 | 0.3548 | 0.2360 | 0.2360 | 0.2498 | 68.85 |
| The deciduous | 1.8443 | 1.4497 | 0.4133 | 0.2711 | 0.2676 | 0.1627 | 84.43 |
| The evergreen | 1.9754 | 1.4742 | 0.4455 | 0.2891 | 0.2854 | 0.2000 | 97.54 |
| Species level | 1.9918 | 1.4974 | 0.4630 | 0.3016 | 0.1834 | 0.1943 | 99.18 |

(Fig. 2E): ZYL, TML, WYL, and DYL (purple); WYC2 (green); WYC1, DYC, and some WYC4 (blue); and the remaining WYC4 and WYC3 (yellow). The samples in the purple groups are deciduous *E. japonica*, and the samples in the green, blue, and yellow groups are evergreen *E. japonica*. Interestingly, the great isolation of population ZYL is not reflected in an unusual level of genetic distance (Figs. 2A and 2E). However, this population forms a distinct entity in the dendrogram. These results showed that the genetic differentiation between different geographic populations is not obvious. Furthermore, from Fig. 2E, there was little genetic infiltration between the evergreen and deciduous populations. The lower average gene flow (*Nm*) of *E. japonica* ($Nm = 0.7804$) also confirms the previous results (Table 2). In addition, we found that the deciduous ($Ne = 1.450$, $H = 0.271$, $I = 0.413$, $PPB = 84.43\%$) and evergreen *E. japonica* had rich genetic variation. ($Ne = 1.474$, $H = 0.290$, $I = 0.446$, $PPB = 97.54\%$) (Table 3).

## Genetic differentiation

The Dice's genetic similarity coefficient ranged from 0.729 to 0.938. Two populations had relatively high Dice's similarity coefficients, indicating a close genetic relationship and small genetic differences between the two populations. The Dice's similarity coefficients were higher between WYC1and WYC2 (0.954), WYC2 and WYC4 (0.921), WYC3 and WYC4 (0.936), and DYL and ZYL (0.938), while the Dice's similarity coefficients between DYC and TML (0.729), DYC and ZYL (0.793), WYL and TML (0.733), WYL and DHC (0.764), and WYL and WYC1 (0.766) were lower (Table 4). Genetic differentiation coefficient (*Fst*) values greater than 0.25 indicates greater genetic differentiation across populations (Wright, 1978). The *Fst* values for WYC1 and WYC2 (0.0883), WYC1 and WYC4 (0.2249), WYC2 and WYC4 (0.2070), WYC2 and WYC3 (0.2350), WYC3 and WYC4 (0.1509), and WYC3 and DYL (0.2172) populations were less than 0.25, and the *Fst* values for other populations were all greater than 0.25 (Table 4). Overall, there was a high similarity coefficient and a low *Fst* value across deciduous or evergreen populations in the same mountain, and there

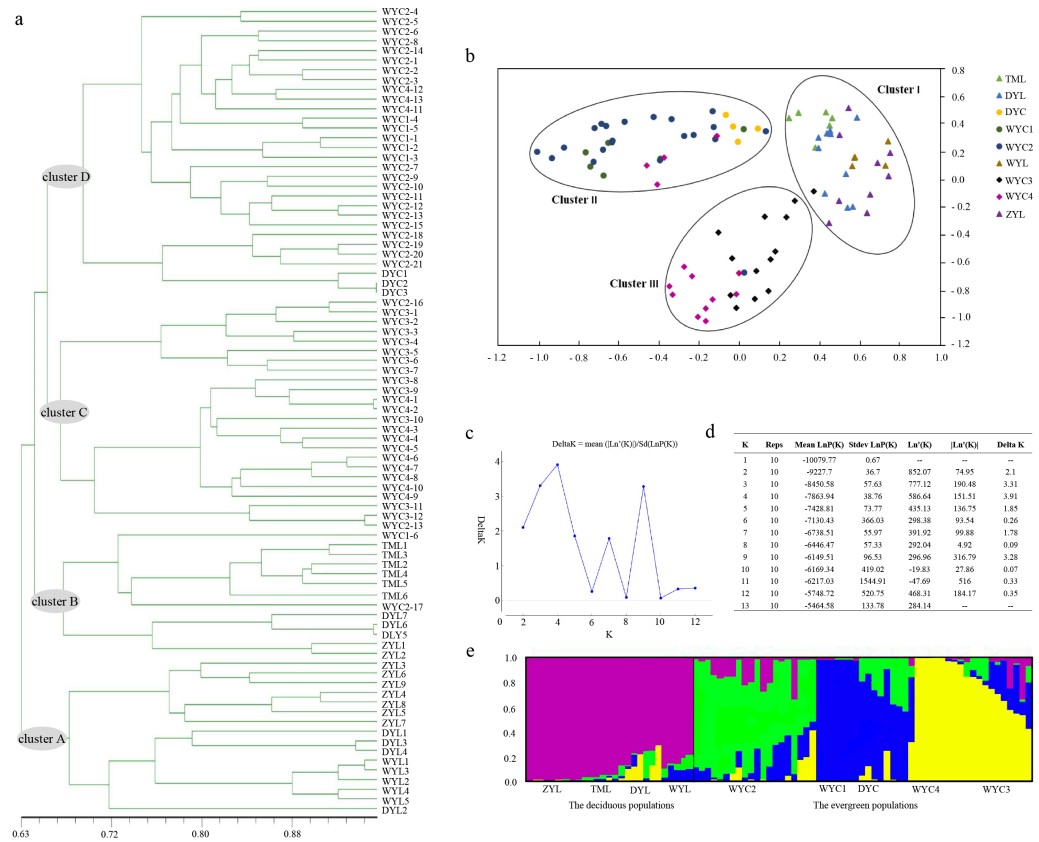

**Figure 2   *E. japonica* population genetic structure.**

was a lower similarity coefficient and high *Fst* value between deciduous and evergreen populations.

The AMOVA analysis showed that 30% of total genetic variation existed across populations and 70% existed within populations (Table 5). The genetic variation within deciduous and evergreen populations accounted for 64% and 77% of the total variation, respectively. The genetic variation across different deciduous and evergreen populations accounted for 36% and 23% of the total variation, respectively. These results showed that for both deciduous *E. japonica* and evergreen *E. japonica*, most genetic variation was from within populations rather than from among populations.

## DISCUSSION

### Genetic diversity and differentiation

We determined polymorphisms in nine *E. japonica* populations from China by estimating the genetic variability across different populations. Compared to previous reports, *E. japonica*' s average *Ne*, *H*, and *I* values ($Ne = 1.4974$, $H = 0.3016$, $I = 0.4630$) were higher than those observed in *Staphylea bumalda* (*Staphylea*, Staphyleaceae) ($Ne = 1.341$, $H = 0.227$, $I = 0.370$) (*Chen, Wang & Wang, 2014*). These results indicate that *E. japonica*

**Table 4** *Fst* value and the genetic similarity coefficient between populations.

| | | TML | DYL | DYC | WYC1 | WYC2 | WYL | WYC3 | WYC4 | ZYL |
|---|---|---|---|---|---|---|---|---|---|---|
| *Fst* | TML | 0.0000 | | | | | | | | |
| | DYL | 0.3600* | 0.0000 | | | | | | | |
| | DYC | 0.6231 | 0.3040 | 0.0000 | | | | | | |
| | WYC1 | 0.4343* | 0.2879 | 0.4110 | 0.0000 | | | | | |
| | WYC2 | 0.3528* | 0.2563* | 0.2740* | 0.0883 | 0.0000 | | | | |
| | WYL | 0.6594 | 0.3379 | 0.6799* | 0.5051 | 0.4236* | 0.0000 | | | |
| | WYC3 | 0.3640* | 0.2172* | 0.3587* | 0.2520* | 0.2350* | 0.3927* | 0.0000 | | |
| | WYC4 | 0.4200* | 0.2921 | 0.4087* | 0.2249 | 0.2070* | 0.4799 | 0.1509* | 0.0000 | |
| | ZYL | 0.3756* | 0.2109 | 0.4042 | 0.3537* | 0.2977* | 0.4002* | 0.2672* | 0.3081 | 0.0000 |
| GS | TML | 1.000 | | | | | | | | |
| | DYL | 0.839 | 1.000 | | | | | | | |
| | DYC | 0.729 | 0.872 | 1.000 | | | | | | |
| | WYC1 | 0.816 | 0.853 | 0.832 | 1.000 | | | | | |
| | WYC2 | 0.861 | 0.912 | 0.883 | 0.954 | 1.000 | | | | |
| | WYL | 0.807 | 0.913 | 0.864 | 0.886 | 0.922 | 1.000 | | | |
| | WYC3 | 0.811 | 0.888 | 0.837 | 0.910 | 0.921 | 0.936 | 1.000 | | |
| | WYC4 | 0.764 | 0.849 | 0.733 | 0.776 | 0.807 | 0.821 | 0.786 | 1.000 | |
| | ZYL | 0.793 | 0.938 | 0.879 | 0.841 | 0.904 | 0.902 | 0.887 | 0.826 | 1.000 |

**Notes.**

*the difference is significant. GS means the Dice's genetic similarity coefficient.

**Table 5** The hierarchical analysis of molecular variance (AMOVA) examining differences among and within populations of *E. japonica*.

| | Source | df | SS | MS | Est. Var. | % |
|---|---|---|---|---|---|---|
| All | Among populations | 8 | 605.564 | 75.696 | 6.455 | 30 |
| | Within population | 79 | 1167.117 | 14.774 | 14.774 | 70 |
| | Total | 87 | 1772.682 | | 21.229 | 100 |
| The deciduous | Among populations | 3 | 205.99 | 68.641 | 7.538 | 36 |
| | Within population | 26 | 351.844 | 13.532 | 13.532 | 64 |
| | Total | 29 | 557.767 | | 21.070 | 100 |
| The evergreen | Among populations | 4 | 258.227 | 64.557 | 4.552 | 23 |
| | Within population | 53 | 815.273 | 15.383 | 15.383 | 77 |
| | Total | 57 | 1073.500 | | 19.935 | 100 |

has high genetic variation at the species level. There are two types of *E. japonica*, deciduous and evergreen *E. japonica*. We evaluated the genetic diversity of *E. japonica*, deciduous *E. japonica*, and evergreen *E. japonica*, and found that they all have abundant genetic variation, and the genetic variation within the population was much greater than that between populations. Numerous previous studies have shown that most woody species have more variation within individual populations than across different populations (*Hamrick, Godt & Sherman-Broyles, 1992*; *Wei et al., 2012*). Woody species with long-lived, outcrossing breeding systems and animal seed dispersal have higher variations within

populations than across populations (*Albert, Raspe & Jacquemart, 2005*; *Schaal et al., 2010*; *Ramírez-Valiente, Valladares & Aranda, 2014*). *E. japonica* is a long-lived plant with a mixed mating system. Bee is the primary pollinator for *E. japonica*, meaning that pollen transfer can occur between adjacent populations (*Sun et al., 2017*), such as the WYC1 and WYC2 populations, WYC3 and WYC4 populations located in the Wuyi Mountain. The red pericarp of *E. japonica* attracts birds to eat the fruit that helps with seed dispersal. The lower interpopulation genetic diversity of *E. japonica*, deciduous *E. japonica*, and evergreen *E. japonica* and higher intrapopulation genetic diversity may be the results of its long life, mixed mating, and seed dispersal systems.

The lower Dice's similarity coefficient and higher *Fst* value between the deciduous and evergreen populations confirmed that the deciduous and evergreen *E. japonica* experienced differentiation. Frequent gene flow can prevent genetic drift and reduce genetic differentiation (*Tremblay & Ackerman, 2001*). The STRUCTURE analysis and the lower average gene flow indicated that there was less gene flow between the evergreen and deciduous populations. According to our long-term field observations, the florescence of deciduous *E. japonica* was observed from April to May, and the florescence of evergreen *E. japonica* was observed from May to June (*Sun et al., 2017*; *Sun et al., 2019*). Although bees are the primary pollinators, there is a small possibility of genetic interactions between the deciduous and evergreen populations in the same Mountain, such as DLY and DYC populations in Daiyun Mountain, and WYC1 and WYL populations in Wuyi Mountain. We speculate that little gene exchange between deciduous and evergreen *E. japonica* may be the result of the differentiation between these two type populations.

## Implications for *E. japonica* conservation

The deciduous *E. japonica* is mainly distributed in southern China, Japan, Korea, and the evergreen *E. japonica* is distributed in Fujian, Jiangxi, Guangxi provinces in southern China (*Li et al., 2018*; *Sun et al., 2019*). According to our field observations, the influence of human activities has been the main challenge affecting the survival of *E. japonica* populations (*Sun et al., 2019*). Therefore, based on the genetic diversity information of the population, an effective *E. japonica* conservation strategy is proposed to create favorable conditions for the innovation and effective utilization of *E. japonica* germplasm resources. We detected the highest genetic diversity in the WYC2 population ($PPB = 86.07\%$, $H = 0.2607$, $I = 0.3971$), followed by WYC3 ($PPB = 70.49\%$, $H = 0.2512$, $I = 0.3751$). The WYC2 and ZYL populations were located in the Jiangshi Nature Reserve and the Xishui Natural National Reserve, respectively, and the DYL population was located in sparsely populated natural villages. These habitats are well-preserved without frequent human interventions. Based on our UPGMA clustering, genetic diversity index, and population location, we suggest that the ZYL, DYL, WCY3, and WCY2 populations in clusters A, B, C, and D should be conserved in-situ. In-situ management of genetic resources can ensure that the majority of extant variation is preserved (*Asddisalem et al., 2016*; *Negri & Tiranti, 2010*).

Furthermore, we should work on recovering populations with low diversity, destroyed habitats, and small population sizes. TML and DYC habitats were shrinking due to human

interventions. The WYC1 population is located near a city, and the WYC3 population was transplanted from local forests to forest farms because of habitat destruction. The populations with the lowest genetic diversity were WYL ($PPB = 20.49\%$, $H = 0.0803$, $I = 0.1174$) and DYC ($PPB = 21.31\%$, $H = 0.0738$, $I = 0.1115$) population. Therefore, we recommend an ex-situ conservation strategy of these populations. In ex-situ conservation, genetic resources are preserved outside their natural habitats in facilities such as seed banks and botanical gardens. These strategies aim to preserve genetic material in collections (*Brush, 2000*; *Negri & Tiranti, 2010*). *Li & Pritchard (2009)* and *Richards et al. (2010)* confirmed that seed collections are a useful and effective way to maintain the size of most ex-situ populations, and that vegetative propagation can rapidly and effectively expand ex-situ population sizes (*Li et al., 2018*). In natural *E. japonica* populations, collecting and storing seeds are of significantly important, especially for populations with low genetic diversity and severely damaged habitats, such as TML, WYL, and WYC1. Other populations with very small individuals, such as DYC, can benefit from vegetative propagation and *in situ* conservation of genetic resources.

## CONCLUSIONS

Our study showed that ISSR is a useful method for characterizing genetic diversity of *E. japonica*, which has high genetic diversity at the species level. UPGMA tree, PCA, and STRUCTURE analysis revealed that *E. japonica* can be divided into deciduous and evergreen *E. japonica*. AMOVA analysis indicated that intrapopulation genetic variation of *E. japonica*, deciduous *E. japonica*, and evergreen *E. japonica* was greater than genetic variation of interpopulation genetic variation, which may due to its mixed mating system and animal seed dispersal. The low average flow (Nm) between deciduous and evergreen populations indicates that there was little genetic infiltration between this two types population, and structural analysis showed also confirmed this result. According to the similarity coefficient and *Fst* value, the deciduous and evergreen *E. japonica* may experience differentiation. We suggested that populations with high genetic diversity of which habitats that are less disturbed by human activities should be protected in-situ, and those populations with low genetic diversity, small populations, and whose habitats are disturbed by human activities should be protected by ex-situ, such as seed collection and vegetative propagation.

### Abbreviations

| | |
|---|---|
| **ISSR** | inter-simple sequence repeat marker |
| **CTAB** | acetyl trimethyl ammonium bromide |
| **UPGMA** | The unweighted pair group method with arithmetic average |
| **N** | north latitude |
| **E** | west longitude |
| **A** | altitude (m) |
| **No. of bands** | the number of amplification band |
| **PIC** | the polymorphism information content |

| *Na* | the number of alleles |
| *Ne* | the number of effective alleles |
| *H* | Nei's genetic diversity |
| *I* | Shannon's information index |
| **No. of polymorphic bands** | the number of polymorphic bands |
| **PPB** | the percentage of polymorphic loci |
| **Nm** | the level of gene flow; except (*He*) and unbiased excepted heterozygosity (*uHe*) |

## ACKNOWLEDGEMENTS

Special thanks to Bin Ou (College of Forestry, Jiangxi Environmental Engineering Vocational College, Jiangxi Province), Zhu-gang Yi (Zunyi Forestry Bureau, Guizhou Province), Zhuang Zou (Zhejiang Academy of Agricultural Sciences), and the Quangzhou Forestry Bureau, who helped us find natural *E. japonica* populations.

### Funding

This work was supported by the Scientific Research Foundation of the Graduate School of Fujian Agriculture and Forestry University (324-1122yb062) and the Special Funds for Leading Scientific and Technological Innovation Talents of Fujian Province (118/KRC16006A). The funders had no role in study design, data collection and analysis, decision to publish, or preparation of the manuscript.

### Grant Disclosures

The following grant information was disclosed by the authors:
Scientific Research Foundation of the Graduate School of Fujian Agriculture and Forestry University: 324-1122yb062.
Special Funds for Leading Scientific and Technological Innovation Talents of Fujian Province: 118/KRC16006A.

### Competing Interests

The authors declare there are no competing interests.

### Author Contributions

- Wei-Hong Sun and De-Qiang Chen conceived and designed the experiments, performed the experiments, analyzed the data, prepared figures and/or tables, authored or reviewed drafts of the paper, and approved the final draft.
- Rebeca Carballar-Lejarazu conceived and designed the experiments, performed the experiments, prepared figures and/or tables, authored or reviewed drafts of the paper, and approved the final draft.
- Yi Yang performed the experiments, analyzed the data, prepared figures and/or tables, authored or reviewed drafts of the paper, and approved the final draft.

- Shuang Xiang and Meng-Yuan Qiu analyzed the data, prepared figures and/or tables, and approved the final draft.
- Shuang-Quan Zou conceived and designed the experiments, performed the experiments, authored or reviewed drafts of the paper, and approved the final draft.

## Data Availability

Raw data are available in a Supplementary File.

## Supplemental Information

Supplemental information for this article can be found online at http://dx.doi.org/10.7717/peerj.12024#supplemental-information.

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
