# Peer review of "Genetic diversity and population structure of *Euscaphis japonica*, a monotypic species"

_PeerJ, doi:10.7717/peerj.12024_

## Round 0.1 · original submission · Major Revisions

This was an invited resubmission of a previously rejected manuscript.

There are many concerns raised by each of the reviewers and I would ask you to address these thoroughly. At this stage, unless the manuscript can be improved substantially it risks rejection at the next round of review.

·

Basic reporting

The study focuses on analysis of genetic diversity and structure of Euscaphis japonica collected from different regions of China using ISSR markers. The data might be used for establishment of conservation program of this species.
The limitations are by use dominant markers as ISSR markers for this study and the use of too limited statistical methods based on similar studies.
The English language is sufficient quality.

Experimental design

Material &Methods section:
-The Methods used are too limited according to the huge works on genetic diversity and structure in woody species.

Validity of the findings

Results section should be revised according with these comments:
-Description of Fst values and similarity coefficient could be moved to “Genetic differentiation” part.
-The UPGMA and PCA analyses should be described in a part entitled “Genetic structure” along with the results of STRUCTURE analysis.
-Results of STRUCTURE analysis should be more described and the genetic diversity parameters of the identified clusters estimated.
Discussion section:
Discussion was too descriptive and should be rewritten. Its no necessary to describe the obtained results in the discussion section but to compare them with results obtained in previous works based on this topic.
I suggest that this study needs more improvement to considered it for publication in this journal.

Additional comments

The study focuses on analysis of genetic diversity and structure of Euscaphisjaponica collected from different regions of China using ISSR markers. The data might be used for establishment of conservation program of this species.
The limitations are by use dominant markers as ISSR markers for this studyand the use of too limitedstatistical methods based on similar studies.
The English language is sufficient quality.
I suggest that this study needs more improvement to considered it for publication in this journal.

Material &Methods section:
-The Methods used are too limited according to the huge works on genetic diversity and structure in woody species.

Results section should be revised according with these comments:
-Description of Fst values and similarity coefficient could be moved to “Genetic differentiation” part.
-The UPGMA and PCA analyses should be described in a part entitled “Genetic structure” along with the results of STRUCTURE analysis.
-Results of STRUCTURE analysis should be more described and the genetic diversity parameters of the identified clusters estimated.

Discussion section:
Discussion was too descriptive and should be rewritten. Its no necessary to describe the obtained results in the discussion section but to compare them with results obtained in previous works based on this topic.

Reviewer 2 ·

Basic reporting

No comment

Experimental design

No comment.

Validity of the findings

No comment.

Additional comments

The manuscript focused on a woody species Euscaphis japonica and revealed its genetic diversity and population genetic structure using ISSR markers. The authors found that E. japonica had an overall high level of genetic diversity and the results of clustering analysis showed limited gene flow between some populations especially between the deciduous and the evergreen populations. Generally, while this manuscript provided some valuable data, the significance of studying the genetic variation of E. japonica and the potential applications of the results are still unclear. In addition, the authors misunderstood the meaning of genetic differentiation and genetic clusters, leading to very confusing results. I therefore think that a major revision is necessary before the acceptance of this manuscript.

Major concerns
1. In the introduction, I suggest the authors to focus on the importance of genetic variation to species persistence especially for endangered species and species with small population size, rather than comparing the genetic diversity of woody species. Moreover, I can’t see why E. japonica must be studied, and the authors should address the reasons why this plant is so important (e.g., endangered species and/or with great potential to be applied in agriculture).
2. In the methods and results, the authors should re-organize their statements in genetic clustering and genetic differentiation. Fst and AMOVA showed the pattern of genetic differentiation among populations, while other analysis like STRUCTURE, UPGMA and PCA allocated individuals into different clusters. I therefore suggest to put the results to the right places.
3. I feel very confused when reading the results of clustering analysis, because it is difficult to conclude whether the clustering patterns obtained from different analyses were converged. For example, whether all individuals assigned into Cluster I in STRUCTURE were allocated into a particular group in UPGMA or PCA. The authors should revise the relevant contents to clearly illustrate their results.
4. The discussion is too redundant. The ISSR is not a prevalent or advanced technique in population genetics, so this part is not necessary. The following two parts are hard to follow without clear topics. I suggest the authors to compare the overall genetic diversity with other related species and then clarify the geographic distribution of genetic diversity with speculations of the underlying mechanisms. As to the genetic differentiation and clustering pattern, the authors may conclude the differentiation pattern at the beginning and discuss the underlying mechanisms from both life history of the plant and the geographic characters. Moreover, in the conservation implications, the authors can discuss which populations ought to have high priority for conservation and provide the global conservation strategy for the studied species.

Minor comment
1. I suggest the authors to provide a sampling map to show the distribution of sampled populations.

---

## Round 0.2 · Minor Revisions

There remain concerns about the logic in this paper that need to be adressed. For instance, in the abstract you say "The genetic variation within the population of E. japonica was much greater than the genetic variation between populations" which is strong evidence to recognise the species as a single entity. You go on to say "We propose that E.
japonica differentiated into deciduous and evergreen populations due to low gene exchange and habitat fragmentation." which fundamentally contradicts the previous statement. Figure 3 suggests there might be three sub-populations, not two within the species and the STRUCTURE analysis suggests four but with overlap. Figure 2 suggests the evergreen plants are derived from the deciduous ones. The STRUCTURE analysis indicates a strong overlap between deciduous and evergreen samples. Arguably the most interesting result here is the lack of correlation between genetic distance and geographic distance. The great isolation of population ZYL is not reflected in an unusual level of genetic distance. However, this population forms a distinct entity in the dendrogram.
Your conservation conclusions are based on your reports of habitat destruction and are at a population level. It is not clear why you would need to differentiate evergreen from deciduous plants in conservation management in that threats are based on geographic position. You should also consider the significance of the Chinese populations in relation to those in Korea and Japan - are these evergreen or deciduous or both? Is the species common outside China and if so, how important is the effort to conserve Chinese populations?

The map you provided uses a copyright image from Google. You need either to redraw this or get permission to use the image. The reviewer suggests there is a need for editing of the English. While most of your manuscript is well written there are areas that could gain from such editing.

Your manuscript has improved considerably since the previous version and I hope that with these further clarifications it will be ready to accept.

Reviewer 2 ·

Basic reporting

No comment.

Experimental design

No comment.

Validity of the findings

No comment.

Additional comments

The authors have addressed all my major concerns, and the manuscript has been improved greatly. Nevertheless, I still have some suggestions: (1) the writing English should be polished by a native speaker, and (2) the authors must plot the sampling map by themselves rather than directly copying the map from Google Earth.

---

## Round 0.3 · Minor Revisions

Thank you for the revisions. Before I accept the submission, Jasmine Janes, the Section Editor, has commented and said:

"Still a few gaps and issues with this one I think.

Several typos still, for example:
Line 209 says 'delta K valve' - this should be value.
Line 216 says 'litter correlation' - I think this should be little correlation

Gilbert et al. 2012 is not the manual for structure, it is a paper presenting guidelines for structure usage in terms of number of iterations to run.

Delta K is not the best indicator of 'optimal' clusters - several recent papers have cited/shown this. It is better to present the log likelihood plot and refer to delta K if needed. The results (last figure) suggest that a higher number of clusters (6-9) would be optimal - they could also use StructureSelector to test the Puechmaille stats. Authors should probably consider substructure too. This figure could be improved. The structure plot should have more distinct bars to separate the populations so readers can assess the number of individuals in each cluster better.

The map (figure 1) is not very good quality. Seems grainy and fuzzy.

Table 5 heading is not very informative. Be clear that this is the AMOVA.

No specific IBD tests performed so the 'correlation' between genetic distance and geographic isolation is not the best terminology for that statement. I understand the intent, and the authors are correct in the presentation that there is little genetic differentiation among geographic populations.

What exactly is the genetic similarity coefficient and what software was used to estimate it? Was it GenAlEx? Is it referring to a Jaccard's S value? Needs more detail AND to be in the methods. The table suggests it is a GS value, but how and where was this calculated?

Could it be clearer on the figures which populations are evergreen vs deciduous considering this is one of the key features here?"

Please address these comments and resubmit so we can move forward.

Regards,

Alastair

Reviewer 2 ·

Basic reporting

No comments.

Experimental design

No comments.

Validity of the findings

No comments.

Additional comments

The authors have addressed all my concerns.

---

## Round 0.4 · Minor Revisions

I have asked a new referee to review your data analysis given comments in previous reviews. There remain some minor corrections to complete.

Reviewer 3 ·

Basic reporting

I found the paper interesting, especially that you can show genetic differentiation supporting your previous phenotypic findings. My comments below are mostly about how you present your results and I hope these could improve the clarity of your manuscript and help your potential readers better understand your results.

Experimental design

I did not see any mention of herbarium voucher specimens in the paper. If you did prepare them, even for a previous paper, they should be mentioned/ cited here as well.

Validity of the findings

no comment

Additional comments

Line 31: I am not sure ‘multiple analyses’ is the best choice here, as the evidence for deciduous and evergreen populations originate in your 2019 paper as far as I can tell. Unless you mean that populations are exclusively either deciduous or evergreen, though I think this would still count as observation, very useful and interesting one, rather than analysis.

Line 32: I don’t think the UPGMA tree can show whether evergreen or deciduous E. japonica originated first. It is interesting that across your analyses these appear as distinct groups, however as the UPGMA analysis cannot be rooted there is no directionality to the tree to assume that deciduous gave rise to evergreen. This could be an interesting further study.

Line 80: I think this should be ‘to date’

Line 80-81: This sentence is unclear

Line 105: I would recommend citing the 2019 paper here, as I assume the field observations refer to that.

Line 199-200: See my comment above about UPGMA

Line 203-204: It would be useful to add that the DYC and WYC populations are evergreen (providing I am correctly interpreting Table 1)

Line 210-211: I am not sure what you mean strong overlap between deciduous and evergreen in the STRUCTURE analysis, especially since in line 215 you mention little genetic differentiation between evergreen and deciduous. It would be useful to add lines to mark populations and evergreen/deciduous in Figure 2e to help the readers understand the results better.

Line 221: What do you mean ‘main populations’?

Lines 228: I think it should be ‘indicates’ instead of ‘indicated’.

Lines 234-235: I find this sentence out of place here. I think it would be better in the discussion to mention that the phenotypic variation established in Sun et al. 2019 is supported by the analyses here and these could suggest that E. japonica is two species rather than one.

Line 258-259: The chance of interpopulation bee pollination would depend on distance between populations. It could be useful to add here how far populations are apart in light of bee foraging distance.

Line 274: I believe this should be ‘little’

Line 294: What do you mean ‘inappropriate habitat’, disturbed by human activity?

Line 306: I think this should be ‘storing seeds’

Line 316: See my comments above about the UPGMA analysis.

Line 320-321: I am not sure what analysis you refer to with Structural analysis. The correlation between genetic and geographic distance (usually a Mantel-test) is only mentioned in relation to population ZYL as far as I can tell. Even then this is more derived from understanding where each population is and to which cluster in the analyses they fall into. It could be useful to include a Mantel-test to show correlation, or lack of, or incorporate geography in the figures if possible.

Line 325: I am not sure, but I think there is no need for ‘in’ in front of ex-situ here.

Table 1. I think it would be useful to add deciduous/ evergreen to the ‘Leaf characters’ column or as an extra column on its own to help the reader understand which population is which.

Looking at the previous comments I see your point about choosing 4 clusters in the STRUCTURE analysis to present comparable results between the different analysis. However, I think in the discussion you should mention how the different number of clusters in the PCA analysis relate to STRUCTURE and UPGMA. Furthermore, you could include the STRUCTURE plot for 6-9 clusters as supplementary information.

---

## Round 0.5 · accepted · Accept

I apologize for the delay; the previous academic editor is no longer available and I have taken over. Thank you for your thorough response to the previous reviews, I am recommending your article for publication.